Journal of Machine Learning Research 23 (2025) 1-10        Submitted  Revised ; Published

# OXA-MISS: A Robust Multimodal Architecture for Chemotherapy Response Prediction under Data Scarcity

**Francesca Miccolis**[1]         FRANCESCA.MICCOLIS@UNIMORE.IT
**Fabio Marinelli**[1]         FABIO.MARINELLI@UNIMORE.IT
**Vittorio Pipoli**[1]         VITTORIO.PIPOLI@UNIMORE.IT
**Daria Afenteva**[2]         DARIA.AFENTEVA@HELSINKI.FI
**Anni Virtanen**[2]         ANNI.VIRTANEN@HUS.FI
**Marta Lovino**[1] ✉         MARTA.LOVINO@UNIMORE.IT
**Elisa Ficarra**[1]         ELISA.FICARRA@UNIMORE.IT
[1] *University of Modena and Reggio Emilia, Italy*
[2] *University of Helsinki, Finland*

**Editor:** TBD

## Abstract

In clinical oncology, tumor heterogeneity, data scarcity, and missing modalities are pervasive issues that significantly hinder the effectiveness of predictive models. Although multimodal integration of Whole Slide Imaging (WSI) and molecular data has shown promise in predicting overall survival (OS), current approaches often struggle when dealing with scarce and incomplete multimodal datasets, a scenario that reflects the norm rather than the exception in real-world clinical practice, especially in tasks like chemotherapy resistance prediction, where data collection is substantially more challenging than for OS.

Accurately identifying patients who will not respond to chemotherapy is a critical clinical need, enabling the timely redirection to alternative therapeutic strategies and avoiding unnecessary toxicity. Hence, this paper introduces OXA-MISS, a novel multimodal model for chemotherapy response prediction designed to handle missing modalities. In the task of chemotherapy response prediction in ovarian cancer, OXA-MISS achieves a 20% absolute improvement in AUC over state-of-the-art models when trained on scarce and incomplete WSI–transcriptomics datasets. To evaluate its generalizability, we benchmarked OXA-MISS on OS prediction across three TCGA cancer types under both complete and missing-modality conditions. In these settings, the results demonstrate that OXA-MISS achieves performance comparable to that of state-of-the-art models. In conclusion, the proposed OXA-MISS is shown to be effective in OS prediction tasks, while substantially improving predictive accuracy in realistic clinical settings, such as the proposed prediction of chemotherapy response. The code for OXA-MISS is publicly available at `https://github.com/AI-BioInformatics/OXA-MISS`.

**Keywords:** chemotherapy response, missing modalities, molecular data, multimodal learning, overall survival, WSI, chemotherapy resistance

## 1 Introduction

Multimodal data integration, particularly of whole-slide imaging (WSI) and molecular data, is increasingly recognized for its potential to enhance clinical outcomes. A surge of models has demonstrated success in predicting overall survival (OS) using these data types (Hou

et al., 2023; Liu et al., 2025; Jaume et al., 2024), leveraging the complementary information captured by histological features and molecular profiles. WSI offers high-dimensional spatial descriptions of tumors, capturing morphological patterns and cellular arrangements (Jaume et al., 2024)(Bontempo et al., 2023). Molecular data, on the other hand, provides a global view of gene expression levels and molecular characteristics within the tumor (Jaume et al., 2024; Barbiero et al., 2020; Lovino et al., 2020). The integration of these modalities promises a more comprehensive understanding of cancer biology and patient prognosis. Nevertheless, existing models often underperform when applied to more clinically complex tasks, such as predicting response to chemotherapy. One of the key challenges lies in the limited availability of data related to very poor or refractory responses to chemotherapy, as well as their complete molecular profiling. While OS is typically well-documented and based on the objectively recorded status of whether a patient is alive at a specific time, chemotherapy response requires expert evaluation on a case-by-case basis, often involving subjective clinical judgment and complex criteria. Consequently, chemotherapy response datasets are generally less common and suffer from limited data availability, making it particularly challenging to apply multimodal approaches that combine molecular data and WSI, as is now standard practice for OS prediction. For instance, the three datasets used for OS prediction in this study include 404, 502, and 397 patients respectively, all with matched WSI and gene expression data. In contrast, for chemotherapy response prediction, our available dataset comprises only 85 patients, of whom just 66 have both modalities. This scarcity is a prevalent challenge in clinical settings, often due to high acquisition costs, limited accessibility, and constraints imposed by specific clinical workflows (Hou et al., 2023; Wu et al., 2024; Xu et al., 2025). As acquiring molecular data can be costly due to the technology and infrastructure required for gene sequencing, especially in underdeveloped areas, the expectation of having full access to complete modalities for integration is not always achievable (Xu et al., 2025). Current methods for multimodal data fusion, particularly with incomplete datasets, struggle to produce reliable predictions for personalized medicine due to the significant heterogeneity in the dimensionality and structure of different medical data types, which makes it inherently difficult to coherently integrate the information. Hence, models capable of operating under missing-modality conditions have recently been developed. ProSurv (Liu et al., 2025) constructs modality-specific prototype banks using intra-modal contrastive learning to capture risk-relevant features, which are then aligned and translated across modalities, with the entire framework optimized via a survival loss. MUSE (Wu et al., 2024) employs a bipartite graph to model patient–modality relationships and addresses modality collapse through mutually consistent contrastive learning, ensuring robust and modality-agnostic representations. HGCN (Hou et al., 2023) leverages individual GCNs to learn intra-modal features, which are aggregated into hyperedges and processed by a Hypergraph Convolutional Network to model complex inter-modal interactions for improved survival prediction. While these models have achieved remarkable performance in OS prediction under missing-modality conditions, they exhibit significant limitations when applied to chemotherapy response prediction in the presence of missing modalities. This highlights a critical gap in the literature: the need for specialized multimodal models capable of effectively handling i) missing modalities in a context of data scarcity and ii) accurately predicting chemotherapy response. Therefore, this paper introduces OXA-MISS (OXA-MISS: Optional Cross-Attention Model for Incomplete Modality

Integration of Whole Slide Imaging and Transcriptomics), a novel multimodal framework designed to flexibly learn from both complete and incomplete modality inputs. By facilitating effective exchange of multimodal information through its optional cross-attention mechanism and aligning both multimodal and unimodal representations within a shared latent space, OXA-MISS enhances predictive accuracy and robustness in both complete and missing modality settings. In particular, we compare our proposed OXA-MISS with state-of-the-art models on a private multimodal ovarian cancer dataset—comprising WSI and molecular data—for chemotherapy response prediction. To mitigate the limitations imposed by data scarcity, this dataset is augmented, in training, with two publicly available WSI-only datasets to enhance the representation of WSIs. The results demonstrate that OXA-MISS outperforms existing multimodal approaches, achieving a 20% absolute improvement in AUC. These findings underscore OXA-MISS's effectiveness in leveraging available data and enhancing predictive accuracy in this clinically important task.

**Contributions.** *i)* We evaluate the performance of state-of-the-art multimodal models designed to operate under missing-modality conditions for chemotherapy response prediction in ovarian cancer; *ii)* we introduce OXA-MISS, a novel multimodal architecture explicitly designed to handle missing modalities, which outperforms existing approaches by achieving a 20% absolute improvement in AUC on the chemotherapy response prediction task in ovarian cancer; *iii)* we demonstrate the generalizability of OXA-MISS by validating its performance on OS prediction across three TCGA cancer types, under both complete and missing-modality conditions.

## 2  Datasets

This section elucidates the datasets employed in this investigation, with a focus on their composition and relevance to the tasks of predicting chemotherapy response and OS.

**Chemotherapy response.** To evaluate this task, we exploited a curated subset from the DECIDER project (DECIDER observational clinical trial - Multi-layer Data to Improve Diagnosis, Predict Therapy Resistance and Suggest Targeted Therapies in HGSC; ClinicalTrials.gov identifier: NCT04846933) and two publicly available ovarian cancer cohorts.

- *DECIDER*: Comprising 324 WSIs obtained at diagnosis from a cohort of 85 ovarian cancer patients undergoing neoadjuvant chemotherapy, the DECIDER dataset includes 29 platinum-resistant and 56 platinum-sensitive cases. The tissue types represented encompass omentum, ovary, peritoneum, adnexa, lymph nodes, fallopian tubes, mesentery, uterus, vagina, and bowel. Treatment response was determined through manual curation based on RECIST 1.1 criteria Eisenhauer et al. (2009). For 66 out of 85 patients, both WSIs and gene expression data are available. This dataset was used for both training and evaluation purposes.
- *PTRC*: it comprises 326 WSIs from 155 ovarian cancer patients (67 refractory, 88 sensitive), incorporating tissue samples from the omentum, ovary, fallopian tubes, vagina, mesentery, uterus, peritoneum, bowel, adnexa, and lymph nodes. Resistance to platinum-based therapy is defined by disease progression observed during or within four weeks following treatment (Chowdhury et al., 2022). This dataset lacks molecular data; therefore, it is exploited for training purposes only.

- *OBR*: 285 WSIs from 78 ovarian cancer patients (43 responsive, 35 resistant to bevacizumab). Only ovarian tissue is included (Wang et al., 2021)). Given the absence of molecular data, this dataset is utilized solely for training purposes.

Overall, the chemotherapy response dataset used in this study comprises 318 patients, of whom 66 have both gene expression and WSI data, while the remaining patients have WSI data only. OXA-MISS is designed to fully exploit this limited yet comprehensive dataset, currently the most complete public and private resource for this task. Given the lack of agent-specific cohorts, the model is trained on multiple chemotherapy modalities to provide a general prediction of response rather than being restricted to a single agent.

**Overall Survival.** For the task of predicting overall survival, we utilized three public datasets from The Cancer Genome Atlas (TCGA) repository, a recognized resource offering a wide array of cancer samples. Specifically, we focused on breast (BRCA), kidney (KIRC), and lung (LUAD) cancers. We selected patients for whom both WSI and gene expression data were available. This stringent selection methodology yielded the following cohort sizes for each cancer type: BRCA (404 patients), KIRC (502 patients), and LUAD (397 patients).

**Preprocessing.** Each slide is processed with CLAM (Lu et al., 2021) to extract non-overlapping patches $p$ of dimensions $256 \times 256$ at up to $20\times$ magnification and encoded with UNI's vision encoder (Chen et al., 2024) to obtain 1024-dimensional feature vectors. For gene expression in the Chemotherapy Response task, we employed a curated set of 82 genes from (Afenteva et al., 2024). This set includes differentially expressed genes between refractory and sensitive patients, as well as genes associated with the JAK-STAT and Hypoxia pathways, selected based on Progeny-derived activity differences. Instead, in the OS prediction task, we used a set of 5 gene groups, comprising a total of 1,742 genes, obtained from MSigDB website (Broad Institute and UC San Diego). The selected gene groups are *Tumor Suppression*, *Oncogenesis*, *Protein Kinases*, *Cellular Differentiation*, and *Cytokines and Growth*. Gene expression values were normalized using a standard z-score transformation computed on the training set. The normalization was applied feature-wise to each gene and propagated to the validation/test sets. Gene expression vectors were then encoded to match the dimensionality of the patient's WSI embedding.

## 3 Proposed Method

We introduce OXA-MISS (OXA-MISS: Optional Cross-Attention Model for Incomplete Modality Integration of Whole Slide Imaging and Transcriptomics), a novel multimodal deep learning architecture specifically designed for the prediction of chemotherapy response. A central innovation of OXA-MISS lies in its optional cross-attention branches, which enable effective integration of Whole Slide Images (WSI) and genomic data when both modalities are available; otherwise, the model seamlessly defaults to unimodal processing. Furthermore, OXA-MISS projects both unimodal and multimodal representations into a common latent space—referred to as the patient embedding in Fig. 1—thereby promoting alignment between the respective modality-specific encoders. This architectural design enhances the model's robustness and adaptability, making it particularly well-suited for real-world clinical settings where data completeness is often inconsistent.

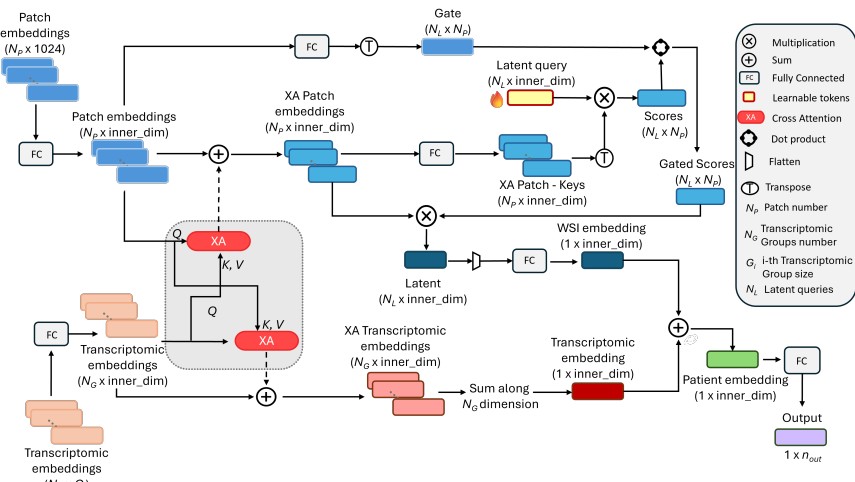

Figure 1: Overview of our proposed OXA-MISS. The grey area indicates the optional multimodal cross attention integration.

## 3.1 OXA-MISS Architecture

As depicted in Fig. 1, our model receives two input branches: the patch embeddings of shape $(N_P, 1024)$ and the transcriptomic embeddings of shape $(N_G, G_i)$. These tensors are projected into a shared representation space through fully connected layers, resulting in representations of dimensions $(N_P, inner\_dim)$ and $(N_G, inner\_dim)$, respectively.

**Optional Multimodal Integration.** When both modalities are available, a cross-attention mechanism is activated to enable bidirectional information exchange between the WSI and transcriptomics branches. Specifically, two cross-attention operations are performed: in one, the patch embeddings from the WSI branch serve as queries while the genomic embeddings act as keys and values; in the other, the roles are reversed. The output of each cross-attention module is then added to the corresponding query modality, facilitating multimodal interaction and alignment. In cases where one modality is missing, the cross-attention module is deactivated and replaced with a neutral tensor, preserving the structure of the computation without introducing additional information. To accommodate scenarios in which only a single modality is available, we define the *optional cross-attention mechanism* as a gated operation:

$$\widehat{X}_m = X_m + \mathbb{I}_{\text{avail}}(X_n) \cdot XA(X_m, X_n, X_n), \tag{1}$$

where $X_m$ and $X_n$ represent the embeddings of the querying and supporting modalities (e.g., WSI and transcriptomics, or vice versa), and $\mathbb{I}_{\text{avail}}(\cdot) \in \{0, 1\}$ is an indicator function that equals 1 if modality $X_n$ is available and 0 otherwise.

Each of the two cross-attention modules is parameterized by four learnable projection matrices $W_Q, W_K, W_V, W_O \in \mathbb{R}^{inner\_dim \times inner\_dim}$. Given a query tensor $Q$, and key–value tensors $K$ and $V$, the cross-attention $XA$ output is computed as:

$$XA(Q, K, V) = \text{Softmax}\left(\frac{(QW_Q)(KW_K)^{\top}}{\sqrt{d}}\right)(VW_V)W_O, \tag{2}$$

where $d = inner\_dim$ denotes the dimensionality used for scaling. This design ensures that in the absence of one modality, the cross-attention module is bypassed and the querying

branch proceeds without alteration, while preserving structural consistency in the model's forward pass. This gating mechanism allows OXA-MISS to seamlessly adapt to both unimodal and multimodal input configurations, enhancing its flexibility in real-world clinical scenarios characterized by heterogeneous data availability.

**WSI Branch.** The WSI branch includes a gating mechanism that selectively minimize the influence of patches considered less informative, thereby enhancing the relevance of the features propagated through the network. Concurrently, a set of learnable latent queries with dimensionality $(N_L, inner\_dim)$ is employed to generate diverse representations of the patch embedding tensor via cross-attention. The gating scores are combined with the cross-attention scores associated with the latent queries via a dot product, resulting in weights that guide the formation of $N_L$ distinct linear combinations of the patch embeddings. The resulting tensor is then flattened into a shape of $(1, N_L \cdot inner\_dim)$ and subsequently projected to a final representation of shape $(1, inner\_dim)$ called *WSI embedding*.

**Transcriptomics Branch.** This branch is responsible for generating the transcriptomic embedding tensor. It is obtained by summing the $N_G$ individual vector representations—each of shape $(1, inner\_dim)$—corresponding to distinct transcriptomics groups. The resulting aggregated tensor has a final shape of $(1, inner\_dim)$.

**Multimodal Representation.** The WSI and transcriptomic embeddings are combined via element-wise summation into a unified patient embedding. In cases where one modality is unavailable, it is replaced with a neutral tensor that does not alter the result of the summation. This approach ensures that unimodal and multimodal inputs are projected into a shared representation space, thereby reducing the risk of the final fully connected layer, responsible for mapping the patient's embedding to the $n_{out}$ output neurons, receiving out-of-distribution inputs. Additionally, this design promotes alignment between the unimodal encoders.

## 4 Experiments

In our experiments, we compare the proposed OXA-MISS with state-of-the-art models for multimodal integration of WSI and transcriptomics, including SurvPath (Jaume et al., 2024), ProSurv (Liu et al., 2025), HGCN (Hou et al., 2023), and MUSE (Wu et al., 2024). In all tables, bold values indicate the best-performing model, while underlined scores denote the second-best. To ensure a fair comparison, we use each model's recommended hyperparameters, learning rate scheduler, and optimizer settings. We evaluate all models on both chemotherapy response and overall survival prediction tasks under missing-modality conditions. The chemotherapy response task is framed as a binary classification problem and assessed using AUC, accuracy, and F1-score. To further test robustness despite the limited availability of chemotherapy response data, we also benchmarked OXA-MISS on the OS prediction task using three independent TCGA cohorts, thereby validating its generalizability across settings. For overall survival prediction, we adopt the formulation used in SurvPath (Jaume et al., 2024) and employ the concordance index (c-index) as the evaluation metric.

For the chemotherapy response prediction task, we perform experiments on our private multimodal dataset (DECIDER), which we augment with two public *unimodal WSI* datasets (OBR and PTRC). To assess the robustness of our approach, we employ a 5-fold cross-

Table 1: Comparison of chemotherapy response prediction in ovarian cancer between state-of-the-art models and the proposed OXA-MISS on DECIDER dataset.

| Metric | SurvPath | HGCN | MUSE | ProSurv | OXA-MISS |
|---|---|---|---|---|---|
| AUC | $0.609 \pm 0.057$ | $0.536 \pm 0.081$ | $\underline{0.629 \pm 0.141}$ | $0.418 \pm 0.066$ | $\mathbf{0.824 \pm 0.067}$ |
| Accuracy | $\underline{0.666 \pm 0.019}$ | $0.599 \pm 0.057$ | $0.614 \pm 0.093$ | $0.616 \pm 0.066$ | $\mathbf{0.752 \pm 0.075}$ |
| F1-score | $0.490 \pm 0.054$ | $0.424 \pm 0.045$ | $\underline{0.543 \pm 0.105}$ | $0.396 \pm 0.048$ | $\mathbf{0.726 \pm 0.078}$ |

Table 2: OS performance comparison with models trained on complete-modality data.

| Test | Dataset | SurvPath | HGCN | MUSE | ProSurv | OXA-MISS |
|---|---|---|---|---|---|---|
| Complete | BLCA | $0.561 \pm 0.077$ | $0.563 \pm 0.086$ | $0.543 \pm 0.030$ | $\underline{0.570 \pm 0.004}$ | $\mathbf{0.593 \pm 0.019}$ |
| | KIRC | $0.644 \pm 0.077$ | $\mathbf{0.734 \pm 0.014}$ | $0.624 \pm 0.075$ | $0.691 \pm 0.012$ | $\underline{0.732 \pm 0.006}$ |
| | LUAD | $0.578 \pm 0.067$ | $\underline{0.585 \pm 0.014}$ | $0.561 \pm 0.070$ | $0.576 \pm 0.003$ | $\mathbf{0.606 \pm 0.019}$ |
| WSI-only | BLCA | N.A. | $0.561 \pm 0.023$ | $0.535 \pm 0.027$ | $\underline{0.572 \pm 0.005}$ | $\mathbf{0.588 \pm 0.024}$ |
| | KIRC | N.A. | $\mathbf{0.724 \pm 0.016}$ | $0.601 \pm 0.079$ | $\underline{0.692 \pm 0.014}$ | $0.652 \pm 0.015$ |
| | LUAD | N.A. | $\mathbf{0.579 \pm 0.013}$ | $\underline{0.576 \pm 0.039}$ | $0.567 \pm 0.004$ | $0.536 \pm 0.027$ |
| Transcript-omics-only | BLCA | N.A. | $\underline{0.556 \pm 0.003}$ | $0.542 \pm 0.010$ | $0.515 \pm 0.007$ | $\mathbf{0.584 \pm 0.028}$ |
| | KIRC | N.A. | $0.602 \pm 0.009$ | $0.599 \pm 0.093$ | $\underline{0.661 \pm 0.004}$ | $\mathbf{0.714 \pm 0.016}$ |
| | LUAD | N.A. | $0.554 \pm 0.008$ | $0.567 \pm 0.071$ | $\underline{0.581 \pm 0.008}$ | $\mathbf{0.594 \pm 0.052}$ |

validation strategy and report the mean and standard deviation for each evaluation metric. Since only DECIDER contains both modalities, we partition it into five hold-out folds, while augmenting the training set in each fold with the public WSI-only datasets. This augmentation increases the number of training samples, albeit unimodal, thereby creating a challenging multimodal learning scenario characterized by a high proportion of missing-modality samples. The results, presented in Tab. 1, show that OXA-MISS outperforms state-of-the-art models by approximately 20% in AUC and 18% in F1-score. All models are trained using the same data splits under the augmented training setting of OXA-MISS, which includes WSI-only patients of both public and private datasets. Notably, SurvPath does not support missing modalities, hence it is instead trained only on complete private samples. In this setting, OXA-MISS demonstrates superior performance by a substantial margin, highlighting its effectiveness in learning under conditions of severe data scarcity and incomplete modality availability. Meanwhile, ProSurv and HGCN underperform compared to SurvPath, with MUSE being the only model capable of surpassing it. This outcome underscores the difficulty even models specifically designed to handle missing-modality scenarios face when learning in low-data regimes.

Furthermore, we conducted the overall survival prediction task on three TCGA cancer types: BLCA, KIRC, and LUAD. Owing to the greater data availability in these cohorts, we included only patients with complete WSI and transcriptomics information in our analysis. We trained the models only with complete samples and evaluated performance under three testing conditions: using complete data, WSI-only (by removing transcriptomics), and transcriptomics-only (by removing WSI). To assess the statistical significance of the results, we employed a 5-fold cross-validation strategy and report the mean and standard deviation of each metric across three different seeds. As shown in Tab. 2, our OXA-MISS

Table 3: Comparison for OS prediction training with a 60% missing-modality rate applied **only during training**, where either WSI or transcriptomics is randomly dropped for each incomplete sample. Testing conditions remain identical to Tab. 2, i.e., evaluation is performed on complete data, WSI-only, and transcriptomics-only.

| Test | Dataset | HGCN | MUSE | ProSurv | OXA-MISS |
|------|---------|------|------|---------|----------|
| Complete | BLCA | **0.575** ± 0.007 | 0.520 ± 0.033 | 0.553 ± 0.011 | 0.562 ± 0.014 |
| | KIRC | 0.671 ± 0.002 | 0.583 ± 0.029 | 0.660 ± 0.008 | **0.683** ± 0.003 |
| | LUAD | 0.545 ± 0.027 | 0.577 ± 0.021 | 0.555 ± 0.027 | **0.642** ± 0.013 |
| WSI-only | BLCA | **0.580** ± 0.007 | 0.529 ± 0.059 | 0.552 ± 0.012 | 0.548 ± 0.007 |
| | KIRC | **0.666** ± 0.007 | 0.617 ± 0.035 | 0.641 ± 0.006 | 0.659 ± 0.005 |
| | LUAD | 0.542 ± 0.018 | 0.546 ± 0.031 | **0.565** ± 0.024 | 0.550 ± 0.019 |
| Transcript-omics-only | BLCA | 0.519 ± 0.013 | 0.526 ± 0.016 | 0.519 ± 0.010 | **0.575** ± 0.013 |
| | KIRC | 0.570 ± 0.012 | 0.580 ± 0.025 | 0.659 ± 0.015 | **0.676** ± 0.019 |
| | LUAD | 0.535 ± 0.023 | 0.530 ± 0.010 | 0.547 ± 0.011 | **0.571** ± 0.023 |

achieves performance comparable to state-of-the-art models in multimodal overall survival prediction under missing-modality conditions. Finally, to further evaluate the models' capabilities on the overall survival task, we repeated the previous experiment by introducing a 60% missing-modality rate during the training phase. Specifically, for 60% of the training samples, one modality—either WSI or transcriptomics—was randomly omitted. As in previous experiments, we employed a 5-fold cross-validation strategy across three different seeds. The results, presented in Tab. 3, demonstrate that OXA-MISS remains competitive with state-of-the-art models under these missing-modality conditions.

## 5 Conclusion

In this work, we presented OXA-MISS, a novel multimodal framework designed to address the challenges of data scarcity and missing modalities in clinical oncology by integrating whole slide imaging and transcriptomic data. OXA-MISS is specifically tailored to operate under incomplete data conditions, making it particularly well-suited for real-world applications. Our experiments demonstrate that OXA-MISS achieves substantial performance gains in predicting chemotherapy response in ovarian cancer, with a 20% absolute improvement in AUC over state-of-the-art baselines. Furthermore, OXA-MISS maintains competitive performance in overall survival prediction across multiple TCGA cancer types, under both complete and incomplete data scenarios. The proposed OXA-MISS is designed to cope with clinically relevant scenarios characterized by data scarcity and incomplete modalities. Nevertheless, we also demonstrated its good performance under wider data availability conditions. In future work, additional datasets will be analysed to further assess its generalizability.

## 6 Funding
This study was funded by the European Union's Horizon 2020 research and innovation programme DECIDER (965193) and the FARD initiative of DIEF Dept. of UNIMORE.

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
