# OpenReview forum: "OXA-MISS: A Robust Multimodal Architecture for Chemotherapy Response Prediction under Data Scarcity"
_MICCAI.org/2025/Workshop/COMPAYL — COMPAYL 2025_

### Official Review · Reviewer_g1RZ · 2025-07-09
**Review - OXA-MISS: A Robust Multimodal Architecture for Chemotherapy Response Prediction under Data Scarcity**

**Rating:** 4
**Confidence:** 3

**Review:**

Summary:
This manuscript outlines a multimodal model for the prediction of chemotherapy response in patients with ovarian cancer. This model demonstrates improved performance (20% AUROC) when compared to state-of-the-art prediction models. The authors neatly use an optional cross attention framework which can be bypassed in the case of missing modalities. The authors further report generalizability of the model in predicting overall survival in cohorts of breast, kidney and lung cancer, with metrics comparable to existing prediction models.
This work presents a novel multimodal framework for predicting chemotherapy resistance in ovarian cancer. However, the abstract of this work is dense, making it difficult for the reader to easily access and understand the importance of this work. The authors would benefit from small typographical edits throughout (outlined below) to further improve clarity.

Strengths:
    • This model demonstrates 20% improved performance in conditions of data scarcity when compared to state-of-the-art models in predicting chemotherapy response.
    • The design of the model with optional cross-attention neatly handles missing data facilitating prediction in these instances.
    • This work provides a framework through which to predict chemotherapy response in ovarian cancer, a finding which may have clinical utility in the future.

Weaknesses:
    • Due to the nature of chemotherapy resistance datasets, the model is trained on a limited number of data points, especially as 5-fold cross validation was used. Training with additional datasets would improve the generalizability of the model.
    • This work would benefit from further external validation of the model on a separate cohort to demonstrate the robustness of the model, further convincing readers of its clinical utility.
    • The authors note different chemotherapy modalities within their cohorts, platinum-based and antibody based (bevacizumab). It would be helpful for the reader if the authors could clarify if the long term goal of the model is to predict resistance for all chemotherapy types, or if this should be specific for different agents, as they are likely to have different mechanisms of resistance. In which case, the authors should further justify the inclusion of their selected training cohorts.
    • Results within tables are inconsistently reported. Table 1 underlines the best performing model (excluding OXA-MISS, which is in bold) whilst Tables 2 and 3 switch between underlining and emboldening the font. This is confusing for the reader and should be addressed.
    • The test categories in Table 3 are confusing and should be modified to “WSI - 60% missing”, “Transcriptomics - 60% missing” or similar.
    • Different font sizes are used within Figure 1.
    • Minor typographical/grammatical errors throughout.

---

### Official Review · Reviewer_zKww · 2025-07-15
**Innovative Multimodal Model for Incomplete Data in Ovarian Cancer Response Prediction, Undermined by Limited Validation and Clarity**

**Rating:** 4
**Confidence:** 4

**Review:**

Short summary:
The paper introduces OXA-MISS, a novel multimodal deep learning model designed to predict chemotherapy response in ovarian cancer, with a particular focus on handling missing and incomplete data modalities. The model goal is to remain robust and accurate even when some modalities are unavailable. In experiments, it achieves improvement over state-of-the-art models for response prediction on limited WSI–transcriptomics datasets.

Strengths :
The paper presents a robust and practical model for multimodal learning in scenarios with incomplete data, addressing a critical real-world challenge where essential modalities (e.g., WSI or transcriptomics) are often missing. A key strength of the work is its  evaluation against state-of-the-art methods, demonstrating superior performance in predictive accuracy. The model’s performance is further validated through testing on both public and private datasets spanning diverse cancer types, underscoring its potential for clinical and research applications.

Weaknesses :
The paper lacks clarity in several areas. Notably, the mathematical formulation describing the model is incomplete, making it difficult to fully understand the underlying methodology. Additionally, the generalisability of the proposed approach is questionable due to the limited scope of experiments. The evaluation is not comprehensive enough to convincingly demonstrate the generalisation of the model across different datasets or settings.

Detailed comments (if applicable):
The model demonstrates improved performance in predicting chemotherapy response—showing gains in AUC, accuracy, and F1 score—this result is reported on a single dataset, limiting the strength of the claim.